# Hfq C-terminal region forms a β-rich amyloid-like motif without perturbing the N-terminal Sm-like structure

Mélanie Berbon[1], Denis Martinez[1], Estelle Morvan[2], Axelle Grélard[1], Brice Kauffmann[2], Jehan Waeytens[3], Frank Wien[4], Véronique Arluison[5,6✉] & Birgit Habenstein[1✉]

Hfq is a pleitropic actor that serves as stress response and virulence factor in the bacterial cell. To execute its multiple functions, Hfq assembles into symmetric torus-shaped hexamers. Extending outward from the hexameric core, Hfq presents a C-terminal region, described as intrinsically disordered in solution. Many aspects of the role and the structure of this region remain unclear. For instance, in its truncated form it can promote amyloid-like filament assembly. Here, we show that a minimal 11-residue motif at the C-terminal end of Hfq assembles into filaments with amyloid characteristics. Our data suggest that the full-length Hfq in its filamentous state contains a similar molecular fingerprint than that of the short β-strand peptide, and that the Sm-core structure is not affected by filament formation. Hfq proteins might thus co-exist in two forms in vivo, either as isolated, soluble hexamers or as self-assembled hexamers through amyloid-reminiscent interactions, modulating Hfq cellular functions.

[1] Univ. Bordeaux, CNRS, Bordeaux INP, CBMN, UMR 5248, IECB, Pessac, France. [2] Univ. Bordeaux, CNRS, INSERM, IECB, UAR 3033, Pessac, France. [3] Structure et Fonction des Membranes Biologiques, Université libre de Bruxelles, Bruxelles, Belgique. [4] Synchrotron SOLEIL, L'Orme des Merisiers, Saint Aubin BP48, 91192 Gif-sur-Yvette, France. [5] Laboratoire Léon Brillouin LLB, UMR12 CEA CNRS, CEA Saclay, 91191 Gif-sur-Yvette, France. [6] Université de Paris Cité, UFR SDV, 75013 Paris, France. ✉email: veronique.arluison@univ-paris-diderot.fr; b.habenstein@cbmn.u-bordeaux.fr

Hfq (host factor 1) is a highly conserved protein found in about two-thirds of bacteria[1]. In Gram-negative bacteria, it is implicated in the post-transcriptional stress response through its RNA chaperone function and modulates mRNA translation using small noncoding RNAs (sRNA)[2,3]. Containing an Sm-like globular domain[4], Hfq assembles into a homo-hexameric ring with structurally unlike surfaces, characterized by an electropositive proximal and a non-polar distal face, both of which can interact with RNA[5–7], as well as the convex rim[8]. Extending outside this Sm-core, Hfq also comprises a variable C-terminal region (CTR) with an unknown structure in the full protein functional conformation (Fig. 1). These ~40-residue C-terminal in *Escherichia coli* Hfq adopts an intrinsically disordered fold in solution[9], and none of the numerous Hfq crystal structures reveals its folding[6,10–12]. Nevertheless, it has been shown that numerous functions could be related to the CTR, such as stabilizing the toroidal hexameric structure[13], sRNA binding[14,15], single-, double-, and quadruple-stranded DNA binding[16–19], nucleoid shaping[20], or membrane poration[21,22]. This can be achieved by adjusting its structure and conferring inter- and intramolecular interactions.

In vitro, Hfq can assemble into filamentous architectures like eukaryotic Sm proteins, including the globular domain in hexameric configuration[23]. In vivo, protein propensity to assemble into higher order polymers containing amyloid structural elements can be associated with deleterious effects, such as in neurodegenerative diseases, or beneficial cellular function as in regulatory or scaffolding processes[24–26]. The Hfq C-terminal conformational adjustments, when confronted with molecular partners, have remained enigmatic, and so has its structure in the Hfq filaments[18,22]. Indeed, the structural determinants located in the C-terminal may contribute to Hfq polymerization and reflect important parameters to regulate the activity of Hfq, possibly also explaining its structural role in nucleic acid interactions, shaping, and competition.

We recently confirmed that a minimal assembling motif of 11-residues (Hfq$_{87-97}$), here denominated Hfq11, is capable of assembling in vitro while fragments excluding the peptide sequence fail to do so[21]. In contrast, Hfq11 does not contribute to the C-terminal nucleic acid binding[27]. We here use magic-angle-spinning (MAS) solid-state nuclear magnetic resonance (NMR) combined with complementary biophysical approaches to shed light on the conformation of the minimal assembling motif Hfq11, isolated or within the context of the full-length Hfq in the assembled filamentous architecture. MAS solid-state NMR constitutes a powerful non-destructive technology to study assembled biomolecules because it provides site-specific information on structures, dynamics, and interactions in the assembled state[28–32]. We complement our analysis using X-ray diffraction, synchrotron

radiation circular dichroism (SRCD), and attenuated total reflectance–Fourier-transform infrared spectroscopy (ATR-FTIR).

Our data indicate the formation of a cross-β architecture by Hfq11 in amyloid-like fibers. The peptides are arranged in a well-folded β-strand spine with low flexibility remaining in the primary peptide structure. We further reveal that full-length Hfq monomers assemble into highly ordered structures with a unique conformation in the polymeric assembly. Surprisingly, segments of the CTR are part of the rigid filamentous core in a clearly defined conformation. MAS solid-state NMR data further indicate that the C-terminal segment in the fibrillar core contains the Hfq11 sequence and that this region adopts a similar β-strand fold as in the Hfq11 fragment in isolation.

## Results

### 11-residue C-terminal Hfq peptide forms filaments with cross-β signature.

We have assembled the minimal Hfq11 motif in water, forming bundles of straight, unbranched fibers of approx. 20 nm (Fig. 2a, b), reminiscent of peptides in amyloid-like fibrils. We then analyzed the fibers by X-ray diffraction, an established technique to detect the presence of a β-amyloid backbone in the powder diffraction pattern of amyloid-like filaments[33–38]. We indeed observe the strongest reflection at 4.6 Å, corresponding to the inter-strand distance between the β-strands along the proto-filament axis (Fig. 2c, d)[39]. The strong hydrogen bonding, typical for the β-amyloid backbone, leads to very repetitive arrangements and explains this intense signal. We observe the typical signal around 7.8–8 Å, which can be assigned to the relatively short inter-sheet distance arising from the tight stacking of the peptide side-chains[36,40]. A more diffuse signal also appears around 3.8 Å. This reflection has been observed multiple times for amyloid-like proteins[36,41,42] and has been attributed to Cα–Cα spacing in the pleated β-strand for fragments of the Amyloid-β (Aβ) protein[35] found in the deposition of amyloid plaques, a hallmark of Alzheimer's disease. In comparison, before washing the sample by centrifugation, the signals at 3.8 Å and 8 Å are doubled (X-ray diffraction shown in Supplementary Fig. 1). The peptide sequence could represent a steric zipper arrangement where multiple interfaces are possible[33,34], explaining the signal doubling if two peptide arrangements are present (Fig. 2d). Since signal doubling mostly disappears after washing and centrifugation of the peptide, they might arise from non-polymerized oligomeric Hfq11 species. In summary, the data indicate that the 11-residue Hfq peptide can indeed assemble into fibers with cross-β architecture.

### 11-residue C-terminal Hfq peptides arrange into amyloid-like β-strand filaments.

The technology of MAS solid-state NMR stands out for its power on amyloid-like protein investigations because it can provide site-specific information on the peptide or protein in an assembled filamentous state. We thus chose to apply MAS solid-state NMR on Hfq11 fibers and obtained a highly resolved $^{13}$C-detected cross-polarization (CP) solid-state NMR spectral fingerprint (Fig. 3a). The signal intensity in the $^{1}$H–$^{13}$C CP, relying on a $^{1}$H–$^{13}$C polarization transfer based on dipolar couplings, shows that the peptide residues are immobilized in the amyloid-like fibrils. The resolution and dispersion of the peptide in the CP (Fig. 3a) shows that the peptide is ordered in structural motifs on the atomic level. Cα and Cβ resonances of proteins are exquisite probes to define the secondary structure of the underlying residue[43]. Because only one threonine (Thr) is present in Hfq11 (SAQN$\underline{T}_{92}$SAQQDS), and the Thr Cβ resonances reveal a unique average chemical shift range in proteins (68–71 ppm)[43], the analysis of the most downfield shifted resonance in the range of Cα and side-chain resonances reveals atomic information on

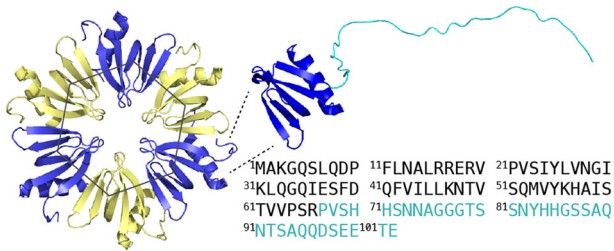

**Fig. 1** *E. coli* **Hfq hexamers (PDB code 3QHS) and monomer predicted by Alphafold[68] (AF-P0A6X3-F1) for *E. coli* Hfq including the C-terminal intrinsically disordered region highlighted in cyan on the structure and on the sequence below.** The Alphafold prediction for the C-terminal region indicates a per-residue confidence score (pLDDT) < 90 for all residues with a residue number > Val 68.

$^{1}$MAKGQSLQDP  $^{11}$FLNALRRERV  $^{21}$PVSIYLVNGI
$^{31}$KLQGQIESFD  $^{41}$QFVILLKNTV  $^{51}$SQMVYKHAIS
$^{61}$TVVPSRPVSH  $^{71}$HSNNAGGGTS  $^{81}$SNYHHGSSAQ
$^{91}$NTSAQQDSEE  $^{101}$TE

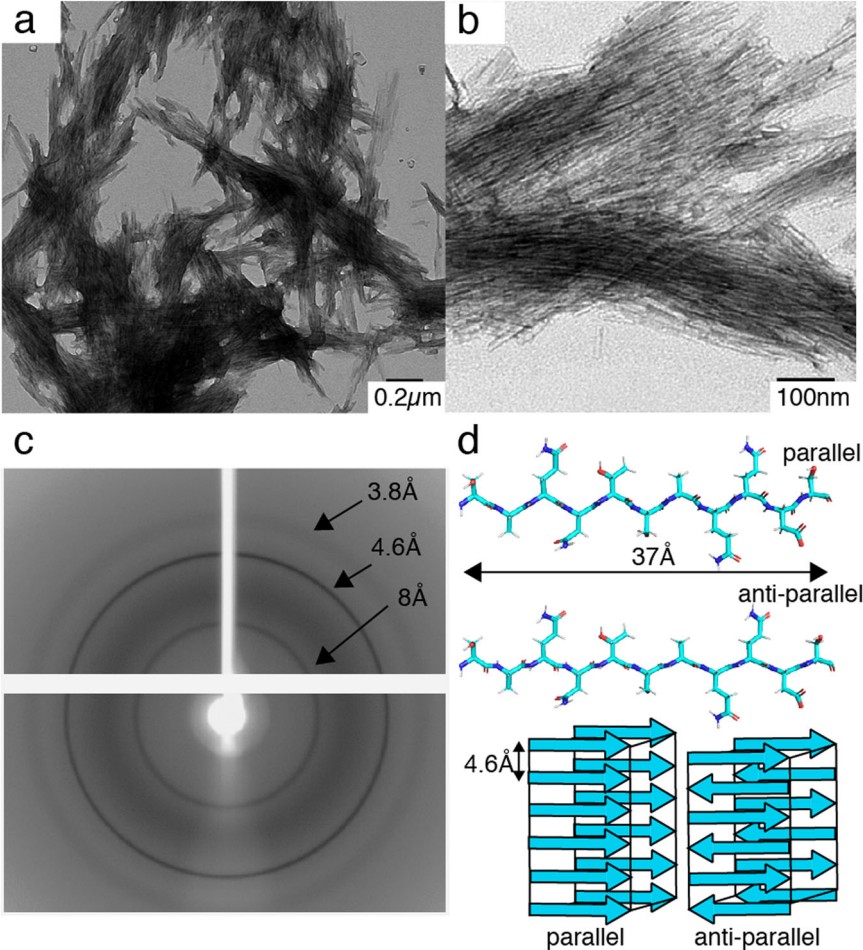

**Fig. 2 Amyloid-like cross-β fibers of synthetic Hfq11 filaments. a, b** Transmission electron micrographs **c** X-ray diffraction pattern. **d** Structural model of a monomeric Hfq11 peptide in a parallel or anti-parallel arrangement (PyMol[69]).

Hfq11 Thr Cβ. When comparing the observed Thr Cβ chemical shift (70.1 ppm) to the average Thr chemical shift in α-helical conformation (68.6 ppm), we can deduce β-strand conformation of the Thr 92 residue (Fig. 3a). Considering the amino acid composition of Hfq11, the Cβ chemical shift of aspartate (Asp) and asparagine (Asn) can further be identified although the secondary structure motif cannot be derived because of the ambiguity to the respective Asp or Asn assignment. The most upfield resonance shifts around 20 ppm represent signals of the alanine (Ala) Cβ in β-strand conformation and Thr Cγ moieties. No signal is detected below 20 ppm, excluding the presence of Ala Cβ in α-helical conformation. The downfield signals ranging from 172.3 to 178.3 ppm correspond to resonances of carbonyl ($^{13}C = O$) moieties in the amino acid sequence (Fig. 3a and Supplementary 2). Signals centered around 178 ppm, highlighted in yellow in Fig. 3a, should correspond to Gln Cδ (3), Asn Cγ (1), and Asp Cγ (1) resonance peaks, whereas backbone carbonyls of a peptide lead to the signals between 172 and 175 ppm. These backbone $^{13}C = O$ resonance shifts clearly indicate a peptide backbone in β-strand conformation because $^{13}C = O$ signals of the amino acids present in Hfq11 should appear upfield of 175.3 ppm (Ala) for a β-strand conformation while arising between 179.5 ppm (Ala) and 176.5 ppm (Ser) if in α-helical conformation[43]. Peak shape and intensity should contain all signals of 11 backbone $^{13}C = O$ when compared to the signal intensity of the 4 side-chain $^{13}C = O$. The resonance peaks at 110–100 ppm correspond to spinning sidebands of the $^{13}C = O$ signals (172–178 ppm) due to the MAS frequency of 11 kHz,

corresponding to approx. 73 ppm at a magnetic field strength of 14.1 T.

To further confirm that Hfq11 is assembled into a uniformly β-strand conformation in the entire peptide, we highlighted the expected average chemical shift pattern of the peptide SAQNT-SAQQDS in β-strand conformation (orange lines) and α-helical conformation (pink lines) on the $^{1}H–^{13}C$ CP Cα and side-chain Carbon chemical shift range (top and bottom panel in Fig. 3b, respectively). Globally, the β-strand peak predictions (orange) correspond better than the α-helical peak predictions to the observed signal pattern. When considering the average signal patterns of the Hfq11 primary sequence in pure β-strand vs. α-helical conformation in detail, several statements can be deduced. (i) In the spectral region of the outmost downfield and upfield resonance shifts, respectively, the Thr Cβ signal is clearly in a chemical shift range, indicating β-strand conformation and no signal of Ala Cβ in α-helical conformation below 20 ppm is visible. (ii) While the signal cluster around 65 ppm can be explained by Cβ signals of the three serines (Ser) in β-strand conformation, the signals would remain unclear if Ser in α-helical conformation were present. Only Thr Cα in α-helical shows a resonance signal at this position and only one Thr is present in the sequence. (iii) The peaks observed around >50 ppm mostly arise from Ala and Asn Cα signals of residues in β-strand conformation and are unexplained if these are in α-helical conformation. (iv) The same is true for Asp and Asn Cβ signals in 45 ppm. (v) Finally, a signal at chemical shift values <30 ppm is missing for glutamine Cβ moieties if the peptide was in α-helical

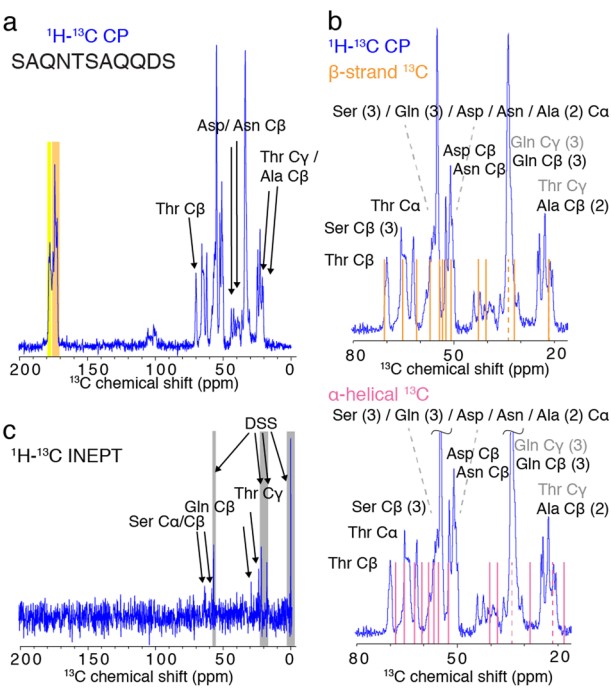

**Fig. 3 $^{13}$C-detected solid-state NMR spectra of Hfq11 on a 14.1 T spectrometer at 11 kHz MAS. a** $^1$H–$^{13}$C cross-polarization (CP) spectrum (blue) detecting signals of atoms in a rigid conformation. In black are annotated unambiguous chemical shift assignments. Highlighted are the spectral regions corresponding to signals of the carbonyl $^{13}$C = O moieties of, respectively, side-chain $^{13}$C of Gln Cδ (3), Asn Cγ (1), and Asp Cγ (yellow) and of the backbone $^{13}$C = O of amino acids in β-strand conformation[43] (orange), see expanded spectrum in Supplementary Fig. 2. **b** $^1$H–$^{13}$C CP spectrum (blue). Orange lines indicate average assignments if the above-annotated corresponding Cα or Cβ atom is in β-strand conformation. Pink lines indicate average assignments if the above-annotated corresponding Cα or Cβ atom is in α-helical conformation[43]. Dotted lines and gray annotations correspond to amino acid side-chain assignments that cannot clearly be related to β-strand or α-helical conformation. **c** $^1$H–$^{13}$C INEPT spectrum (blue) detecting signals of atoms with residual flexibility. In black are annotated unambiguous chemical shift assignments. Signals underlined in gray denote assignments to the reference 4-dimethyl-4-silapentane-1-sulphonic acid (DSS).

conformation. In summary, our spectral analysis strongly indicates that the Hfq11 peptide is present in uniformly β-strand conformation. The weak signals in the $^1$H–$^{13}$C insensitive nuclei enhanced by polarization transfer INEPT spectrum (Fig. 3c) indicate that only some moieties in the amyloid-like assembly are partly mobile. The averaged chemical shifts can be assigned to Ser Cα and Cβ, which are located on both peptide endings, the longer side-chain Thr Cγ as well as Gln Cβ. Strong signals are detected for the soluble referencing compound 4,4-dimethyl-4-silapentane-1-sulphonic acid (DSS).

We then decided to study Hfq11 β-sheet structures by ATR-FTIR and SRCD to determine their arrangements in the amyloid-like structure (Fig. 4). The FTIR spectrum shows three major bands at 1622, 1670 and 1695 cm$^{-1}$ in amide I region (Fig. 4a). The first FTIR band at 1622 cm$^{-1}$ indicates the presence of intermolecular β-sheet structure from amyloid fibrils[44,45]. The 1695 cm$^{-1}$ band is indicative of the antiparallel arrangement of the individual β-strands[46]. As for the amide II region, we also observe a band at 1550 cm$^{-1}$ characteristic for parallel β-strand arrangement[47]. The band around 1740 cm$^{-1}$ corresponds to the C = O stretching vibration of an ester. As active esters are often

used for solid-phase peptide synthesis, we suspect this band coming from slight contamination occurring during peptide synthesis. Since this ester band is located outside of the amide I band, used for the characterization of the secondary structure, it does not interfere with our results. The SRCD spectrum (Fig. 4b) indicates a β-sheet structure with a content of 25% antiparallel and only 5% parallel arrangement confirming the FTIR measurements. In both FTIR and SRCD, the signals of random structure might arise from non-polymerized monomeric or smaller oligomeric peptides. This hypothesis is confirmed by ATR-FTIR measurement since we observe a higher contribution of different conformational states when fibers are deposited on the ATR surface without a subsequent washing step, eliminating non-polymerized monomers (Supplementary Fig. 3). Note that these non-polymerized species are likely not detected in solid-state NMR, due to the centrifugation before rotor filling that selects high-molecular-weight species such as fibers.

**Full-length Hfq filaments contain a rigid globular domain and segments of the CTR.** We then decided to extend our structural analysis by solid-state NMR to full-length $^{13}$C isotope-labeled Hfq fibrils. Note that while isolated CTR self-assembly in the absence of cofactor is usually slow and may take a few weeks[48], the presence of the N-terminal Sm-core accelerates this process. We produced and assembled fully $^{13}$C-labeled Hfq and recorded a $^{13}$C-detected $^{13}$C–$^{13}$C proton-driven spin diffusion (PDSD) MAS solid-state NMR spectrum. This experiment gives principally access to intra-residual $^{13}$C–$^{13}$C correlation signals from $^{13}$C atoms in a rigid conformation (Fig. 5). Signal intensity, resolution, and dispersion of the signals indicate, respectively, an immobile, homogeneously ordered monomer structure containing α-helical and β-strand structural motifs, in the assembled filaments. Based on previous analysis[23], we presumed a conserved monomer globular domain in the structural assembly. We thus predicted the Cα and Cβ resonances for the monomeric Hfq globular domain structure (PDB code 4RCB) using the SPARTA + algorithm[49] and plotted the $^{13}$Cα–$^{13}$Cβ correlation peaks on the spectrum (Fig. 5a). The predicted resonance peaks correspond to the observed correlation peak pattern in the $^{13}$Cα–$^{13}$Cβ spectral region, indicating that the globular domain structure is maintained in the filamentous assembly. To evaluate the sensitivity of chemical shift predictions to structural adaptations of the globular domain, we predicted the chemical shifts for the hexameric Hfq arrangement (PDB code 3QHS). Our data show that the limited structural adaptations of the globular structure between the monomeric and the hexameric arrangement lead to significant modifications of the chemical shift predictions such as for Thr 49, located in the loop connecting β-strand 3 and 4, with a very similar structure in the monomer and the hexamer and a chemical shift difference of 1.7 ppm for Cα as illustrated in Supplementary Fig. 4. We then assigned spin systems that could clearly identify the globular structure and concentrated therefore on isoleucines (Ile), exclusively present in the globular domain. $^{13}$C–$^{13}$C correlation peaks of five Ile spin systems can be assigned (Fig. 5b), colored in a range from red to green with increasing β-strand structural propensity of the underlying residue. Only five Ile are present in full-length Hfq and all detected spin systems display intense signals for all residue atoms, indicating that all five Ile distributed over the Hfq sequence are immobilized in the assembly. The residues are indicated in red on the hexameric crystal structure of the full Hfq (PDB code 3QHS) in Fig. 5c (note that the CTR was not resolved in this structure). The Ile is well-distributed over the primary sequence of the globular domain, indicating that not only parts of the globular structure are visible in the $^{13}$C–$^{13}$C PDSD NMR spectrum. The side chains of Ile30

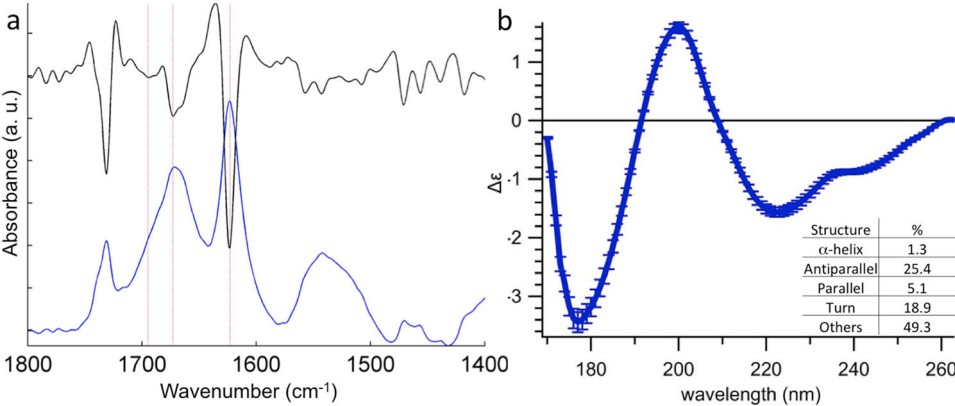

**Fig. 4 Secondary structure of the Hfq11 fibrils by ATR-FTIR and SRCD spectra of Hfq11. a** The ATR-FTIR spectrum in blue and the second derivative in black. The red lines correspond to 1695, 1670, and 1622 cm$^{-1}$ wavenumbers for, respectively, antiparallel, random, and amyloid β-sheet structures. Note also the band at 1550 cm$^{-1}$ in amide II, characteristic of the presence of some parallel β-sheets. **b** The SRCD spectrum and the corresponding secondary structure analysis by Betsel indicate the content of 5% of parallel β-sheet and 25 % of antiparallel β-sheet with an NMRSD of 0.058 for the spectral fitting.

that are oriented towards the hexameric surface seem to be packed in the filamentous assembly so that they can be detected in the spectrum. Peak positions for 4–5 Ala residues are indicated for the $^{13}$Cα–$^{13}$Cβ spin systems (blue), thus revealing the presence of structured parts outside of the globular domain in the filamentous assembly of full-length Hfq because the globular domain contains only 3 Ala. This is confirmed by the signals of 4 Thr spin systems, highlighted by cyan lines, with only two Thr in the globular domain, one exposed to the hexameric surface. This corroborates the assumption that even exposed side chains in the globular domain remain immobilized, based on the rigidified side chains of Ile30.

Since Hfq associates with the nucleic acids[50] but also localizes in the vicinity of membranes[51], we tested the impact of the individual negatively charged lipid component Dioleoylphosphatidylglycerol (DOPG) on C-terminal amyloid stability by FTIR (Supplementary Fig. 5). Over an evolution period of 12 h the fibers disassemble in the presence of the DOPG lipids, indicating the possible modulation of Hfq amyloid assembly by cellular lipid components.

**Hfq11 fibers and full-length Hfq filaments contain β-motifs**. We have shown that the C-terminal domain, encompassing the minimal motif Hfq11, exhibits a propensity to assemble into a cross-β spine arrangement. We further revealed that the full-length Hfq assembly contains intact globular domain structures and stretches of the C-terminal domain in a rigid and well-ordered conformation. Because solid-state NMR signals are very sensitive to the local electrochemical environment of the detected nucleus, depending strongly on the secondary protein structure, we compared the $^{13}$C signal pattern obtained for Hfq11 with the two-dimensional $^{13}$C–$^{13}$C PDSD of the full-length Hfq. All Hfq11 signals could correspond to signals detected in the PDSD of full-length Hfq, as clearly exemplified for the spectral regions of Thr, Ser, and Ala (Fig. 6a). Traces reflecting the Thr signals in the full-length Hfq PDSD with respect to Hfq11 signals demonstrate the striking solid-state NMR signal comparability. Whereas the signals of the Thr Cα–Cβ cross-peak detected at the Cβ resonance frequency with a chemical shift of 70 ppm correspond perfectly to the signals detected in Hfq11, this is not the case for the other Thr as exemplified for the Thr Cα–Cβ cross-peak at the Cβ resonance frequency with a chemical shift of 69 ppm (Fig. 6b). Our data thus indicate that the Hfq11 peptide retains its β-strand conformation in the full-length Hfq assembly.

## Discussion

The respective roles of Hfq's structural components, namely the Sm-like globular domain and the C-terminal intrinsically disordered region, are still unclear. Notably, in *E. coli*, the about 40-residue spanning CTR arranges partially disordered in solution[9] but self-assembles when in isolation[48]. As described above, suggested functions of the CTR include sRNA-based regulation[14,15], DNA binding[16–19], membrane poration[21,22,52], or molecular condensation[53,54].

Interestingly, the evolutionary conservation of the C-terminal amino acid sequence indicates the appearance of two Hfq categories, (1) Hfq proteins of γ- and β-proteobacteria that have an extended C-terminus and (2) other Hfq proteins lacking a CTR, including those of Gram+ bacteria such as *S. aureus* (see sequence alignments[1,27], Supplementary Fig. 6).

We decided to investigate the determinants of *E. coli* Hfq self-assembly using a divide-and-conquer approach at a site-specific level, focusing on the structural implications of the amyloidogenic CTR.

We report the molecular structure of the minimal assembling C-terminal motif of Hfq, the 11-residue peptide denominated Hfq11, spanning from residue 87–97. Removing $Q_{90}N_{91}$ and $Q_{95}Q_{96}$ sequences abolishes the self-assembly, $S_{65}RPVSHHSNNAGGGTT_{80}$ by itself is not able to polymerize[21] and mutating the Serines $S_{88}$, $S_{93}$ or $S_{98}$ in the 11-mer region for an Alanine abolishes the self-assembly[27]. Thus, the length and nature of the sequence are important for the self-assembly[27]. By electron microscopy, we confirm that Hfq11 forms around 20 nm wide filaments. Because our solid-state NMR data indicate that no turn in the monomer structure is present, the 10–20 nm width could reflect the bundling of 4 protofilaments stretching over 37 Å. The solid-state NMR resonance assignment and the X-ray diffraction analysis suggest that Hfq11 assembles into a well-ordered cross-β architecture in the assembled filaments. This amyloid cross-β structure was further confirmed by SRCD and FTIR and determined with a mixture composed mainly of anti-parallel conformation. Considering that this structural segment does not seem to contribute to RNA interactions[55] or DNA binding[27], its propensity to assemble into β-strand amyloid-like arrangements indicates a potential regulative role through self-interactions.

Solid-state NMR data reveal that full-length Hfq filaments contain the globular Sm-domain in a homogeneously ordered unique structure. Our data reveal segments of the CTR in an equally ordered homogeneous structure in the full-length

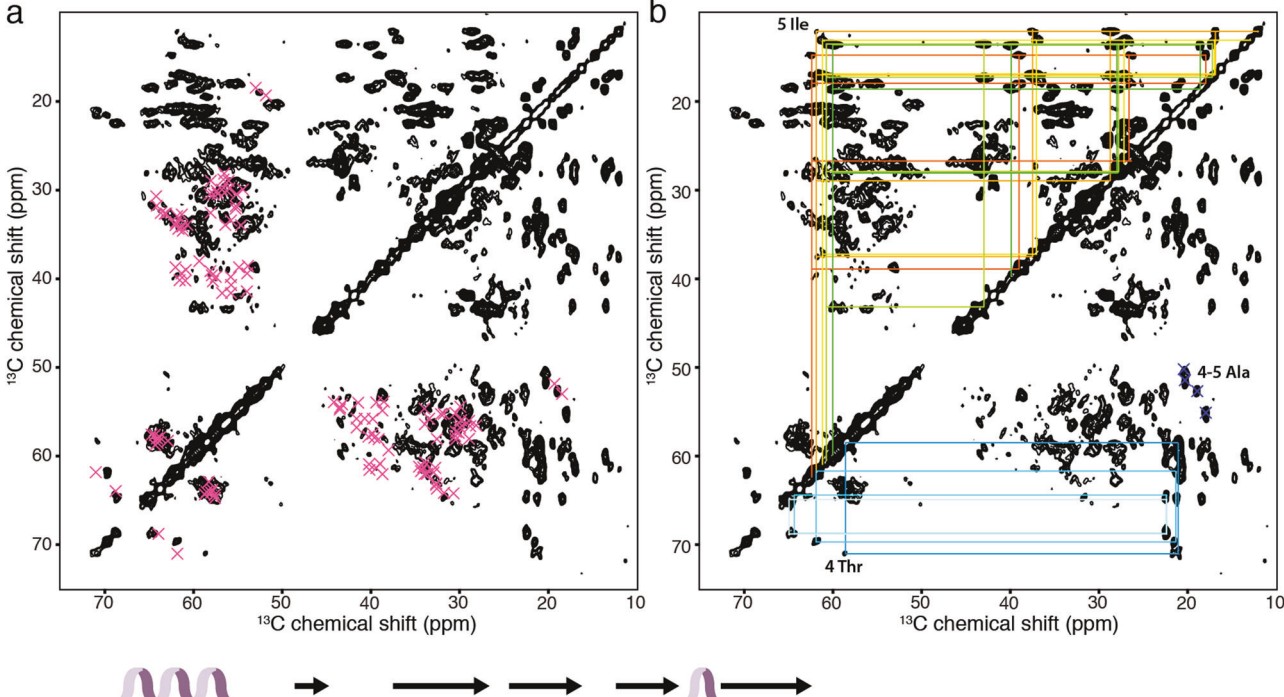

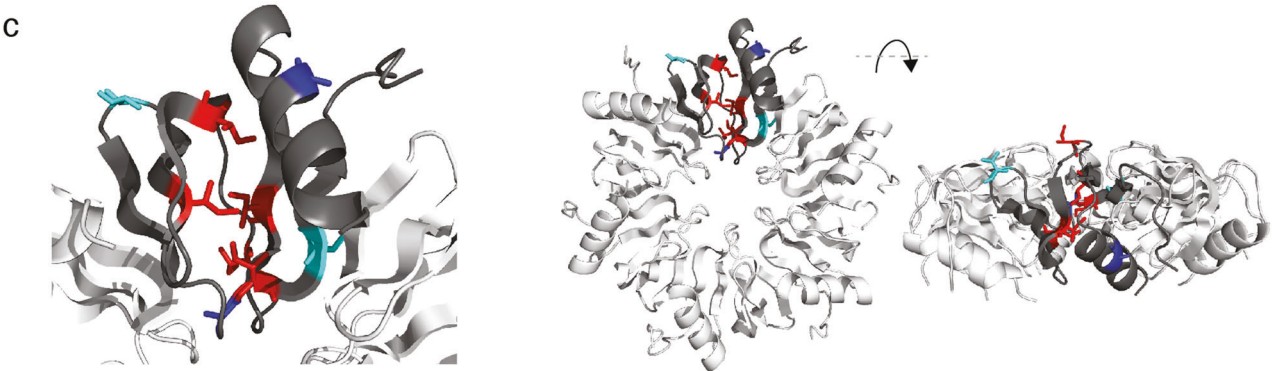

**Fig. 5 Solid-state NMR (ssNMR) analysis of full-length Hfq fibers. a, b** Aliphatic region of a two-dimensional $^{13}$C-detected proton-driven spin diffusion (PDSD) ssNMR spectra of Hfq fibers at 11 kHz MAS on a 14.1 T spectrometer (50 ms mixing time). **a** Resonance peaks (violet) of Hfq monomer structure (PDB code: 4RCB, chemical shift predictions from Sparta+[49]), lacking the last 30 residues, on Hfq fiber spectrum, **b** on the left side of the diagonal, 5 Ile spin system chemical shift assignments ranging in rainbow colors from green to red (increasing β-strand propensities of the Ile residue) are indicated, on the right side 4 Thr assignments in shades of cyan (increasing β-strand propensities from light to dark cyan) and 4–5 Ala assignments (blue) are shown on the spectral fingerprint. Hfq primary and secondary sequence and amino acids corresponding to the assigned spin systems are shown below in the corresponding color scheme (red—Ile, cyan—Thr, blue—Ala). **c** Amino acids of assigned spin systems are depicted in sticks (color scheme as in (**b**)) on the hexameric Hfq structure, lacking the last 30 residues, with a view on the proximal face and the rim (PDB code: 3QHS). Sparta+ chemical shift predictions for PDB code 3QHS and 4RCB are provided in Supplementary Data 1 and 2, respectively.

filaments, with β-strand secondary structure propensity. The assignment of the spin system signals arising from the CTR cannot be simply attributed to a specific segment because the underlying amino acids are present in multiple segments of the CTR. When comparing the Hfq11 signal pattern with the full-length Hfq spectral fingerprint, the Hfq11 signals are found in the Hfq filaments pointing to a contribution of intermolecular interactions and immobilization of these residues. Previously reported as unstructured in solution, we here provide evidence for a structuration of the C-terminal domain in full-length Hfq. Solution NMR had shown that the N-terminal globular domain is conserved within soluble Hfq containing the CTR in a partially disordered conformation[9]. However, the soluble form of full-length Hfq shows detectable conformational differences when compared with the globular domain in isolation. This is, for

example, the case for the residues preceding the segment 65–70, which is capable of folding back onto the proximal face of the hexamers (PDB code 1HK9)[56]. Soluble Hfq might therefore contain the globular domain within a similar structure as described in crystalline hexamers. However, also in the soluble state, Hfq adopts an adapted conformation with parts of the CTR in a not fully disordered state and, therefore, undetectable by solution NMR methods. Our data indicate that the structured C-terminal segments of the full-length Hfq might contain residues 87–97, adopting a tightly packed β-strand conformation in assembled filaments.

In conclusion, here we show that Hfq11 assembles into highly structured amyloid filaments that arrange into stacked β-strands on the atomic level. Furthermore, we have evidence that full-length Hfq contains a β-strand motif in its CTR, reminiscent of an

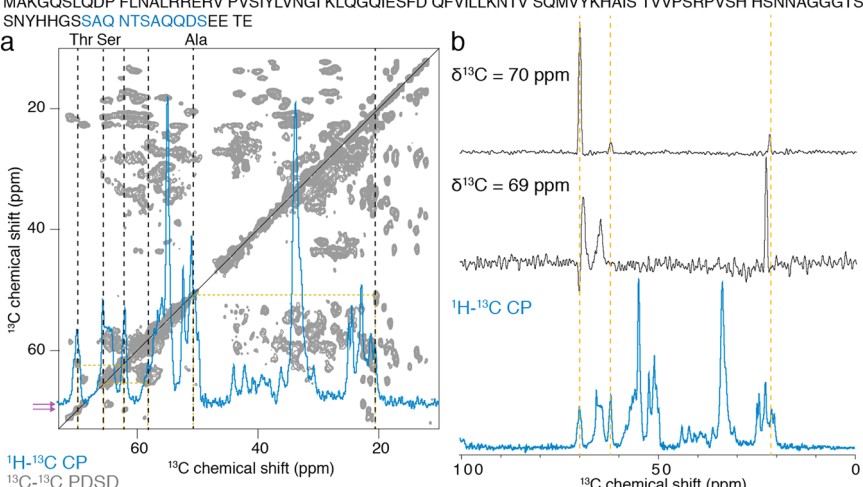

**Fig. 6 Comparison of MAS solid-state NMR analysis of Hfq11 (blue) vs. full-length Hfq (gray).** The full-length Hfq primary sequence is listed in black, with the Hfq11 segment highlighted in blue. **a** Overlay of the $^1H$–$^{13}C$ CP spectrum of Hfq11 with the aliphatic region of the $^{13}C$–$^{13}C$ PDSD of full-length Hfq. Dotted black lines indicate Cα and Cβ signals of Thr, Ser, and Ala residues detected in Hfq11, and dotted yellow lines correspond Cα–Cβ cross-peaks. Potential assignments are plotted above the spectrum. Pink arrows on the y-axis of the PDSD spectrum indicate the trace position shown in (**b**). **b** Traces of two Thr signals in the $^{13}C$–$^{13}C$ PDSD of full-length Hfq plotted above the $^1H$–$^{13}C$ CP spectrum of Hfq11. Yellow dotted lines indicate the corresponding Thr signals.

amyloid conformation and that the β-strand structure forms without perturbing the N-terminal Sm-core. We previously reported that in vitro, Hfq can assemble into filamentous architectures, including the globular domain in a hexameric configuration, and reveal signals that can possibly be attributed to amyloid formation[23]. Our data now indicate that this assembly contains the 11 amino-acid residues in the CTR, allowing the packing of Sm tori through interactions also driven by the CTR. This goes in line with previously observed FTIR data indicating a potential amyloid-like β-sheet motif in full-length fibers[23,57]. Interestingly, the presence of the N-terminal region accelerates the process of self-assembly compared to isolated CTRs[48]. The in vitro observations of oligomeric assemblies containing the CTR and well-folded hexamers suggest that the interactions promoting the molecular assembly might arise transiently in cellulo[58] without disturbing the Sm-core structure and functions while still possibly regulating the activity by steric occlusion. Our data combined with the previous electron microscopy reconstructions and the increase in β-sheet content in Hfq assemblies[23] lead us to suspect a filament architecture containing amyloid-like structural elements and the conserved toroidal rings, schematized in Supplementary Fig. 7. The filament has a helical diameter of 170 Å in which the globular domains of Hfq (illustrated in Supplementary Fig. 7 as blue plates representing the hexagonal monomer arrangements) form repetitive hexameric architectures, leaving a space of approximately 60 Å inside the filament. This interior space could accommodate several arrangements of the CTR, such as a single polymerized intermolecular β-sheet or two complementary β-sheets in an intermolecular amyloid zipper[34,59,60]. We therefore propose the CTRs (here in red) to interact together to provide a backbone architecture, allowing the formation of Hfq filaments. This observation is in agreement with the recent observation that flexible CTR extensions of Hfq form a dense network[61]. These non-covalent CTR amyloid-like interactions could also reflect the conformations adopted in the recently reported single-component and multiple-component elongated Hfq assemblies[53]. Since each hexamer harbors six CTRs (not all represented in the self-assembly of hexamers in Supplementary Fig. 7), the CTRs pointing outside of the filament core cannot contribute to an interior amyloid-like assembly. CTRs, therefore, possibly further stabilize monomer interactions in the hexamer, for instance, via the acidic tip at the end of each CTR interacting with basic patches inside the Hfq globular domain[61] or interconnecting outside the filament core. The observed intrahexamer interactions could potentially be maintained while still accommodating amyloid-like self-assembly along the filament axis. Note that Hfq filaments cannot form in the absence of the CTR, illustrating the central role of this region to promote full-length Hfq self-assembly[23].

Since Hfq has further been found to drive molecular condensation[53,54], to adopt an amyloid-like structuration in cells[58] and to compact DNA depending on the C-terminal Hfq region[20,53], we propose that the here-presented amyloid structuration might govern these local inter-molecular interactions. The propensity of Hfq to assemble into amyloid-like structures and the stability of the resulting assemblies could depend on its cellular localization and the local physico-chemical environment. For example, we here show that the presence of lipids such as DOPG modifies the capacity of the CTR to form amyloid structures. Furthermore, the direct interaction of Polyphosphates (PolyP) with the CTR promotes Hfq condensation[53] and could thus induce amyloid formation; possibly by interacting with the two positively charged Histidine clusters preceding Hfq11. Indeed, PolyP's have been associated with amyloid-inducing effects for disease-related and functional amyloids such as α-synuclein and CsgA[62,63]. The proposed intermolecular interactions between Hfq and PolyP and the here-described amyloid nucleation could provide a structural perspective on the biological regulation of heterochromatin formation in bacteria governed by PolyP-Hfq-DNA[53] interactions.

Moreover, the fact that the eleven-residue sequence contains four residues that can be phosphorylated (three Ser and one Thr) and one residue whose charge state is sensitive to pH (Asp), could further suggest a regulative mechanism of amyloid formation. While Hfq phosphorylation has to our knowledge, not yet been reported, this or other posttranslational modifications, such as acylation[64], could be relevant to regulate its functions.

## Materials and methods

**Protein expression, purification, and assembly.** *E. coli* BL21-DE3 were transformed with a *pTE607* vector containing the DNA

encoding for Hfq[13] and plated onto LB-agar plates containing 100 µg/mL ampicillin. A pre-culture of 100 mL LB medium was inoculated with a single transformed colony and incubated at 37 °C overnight (about 20 h). 1 L of LB medium is inoculated with the pre-culture at OD600 = 0.2 and incubated at 37 °C until OD600 = 0.7–0.8. Protein production was induced with 0.75 mM of IPTG at 30 °C for 20 h. Cells were pelleted at 6000$g$ for 30 min at 4 °C and resuspended in 10 mL lysis buffer (500 mM NaCl, 20 mM Tris-HCl pH 8, 10% Glycerol) containing protease inhibitors (Complete, Roche). Cells were sonicated on ice at 30% magnitude three times (30 s on, 30 s off) and centrifuged at 15,000$g$ for 30 min at 4 °C. The pellet was resuspended in lysis buffer (500 mM NaCl, 20 mM Tris-HCl pH 8, 10% glycerol) containing 1% Triton X-100 and incubated overnight at 4 °C before centrifuging at 250,000$g$. The supernatant was incubated at 80 °C for 15 min, followed by an incubation on ice for 3 min, and centrifuged at 15.000$g$ for 15 min at 4 °C. The supernatant was incubated at room temperature after adding RNase (10 µg/mL) and DNase (50 U/mL) for 1 h. Purification of Hfq was achieved with an Akta Pure 25 HPLC system (GE Healthcare) on a HisTrap affinity column equilibrated in Buffer A (20 mM Tris-HCl pH 8, 300 mM NaCl, 10 mM imidazole). The supernatant was added to the column and washed with Buffer A to remove non-specifically bound proteins. The protein was eluted by increasing the concentration of imidazole in three steps; 25 mM, 50 mM, and 100 mM. As shown in Supplementary Fig. 8, Hfq hexameric species are visible on the SDS gel after purification and therefore resist SDS denaturation. Once the purification was performed, 5 mM EDTA was immediately added to the solution to avoid precipitation of protein. After purification, proteins were buffer-exchanged against a reconstitution buffer (50 mM Tris-HCl, 50 mM NH₄Cl, 10% glycerol, pH = 7.4) using dialysis. Proteins were left to spontaneously assemble at a concentration of 4 mg/ml for 1 week at room temperature under mild agitation. Assembled Hfq was centrifuged and filled into the ssNMR rotor. Hfq 11-residues (SAQNTSAQQDS) and Hfq 38-residues (SRPVSHHSNNAGGGTSSNYHHGSSAQNTSAQQDSEETE) peptides were chemically synthesized (Proteogenix) and assembled in water at 4 °C at 20 mg/mL for >1 month. When needed for the experiment (FTIR), H₂0 would be replaced by D₂0.

**X-ray diffraction.** The diffraction patterns were measured at 270 K on a Rigaku FRX rotating anode X-ray generator at the copper wavelength (Kα, $\lambda = 1.54$ Å). The source is equipped with Osmic Varimax HF optics and a Rigaku© HyPix6000 detector on a 2θ arm of a Rigaku partial chi AFC11 goniometer. The samples were mounted in MicroLoops from MiTeGen on a goniometer head under the cryostream nitrogen flux. The diffraction patterns correspond to a 360° rotation along the phi axis (perpendicular to the direct beam with omega and chi axes at the 0 position) with an exposure time of 360 s. Data were integrated with CrysalisPro (Rigaku Oxford Diffraction, Ltd., Yarnton, Oxfordshire, England). After performing the first X-ray diffraction experiments, we used the remaining sample of Hfq11 filaments, resuspended the filaments in H₂O, and centrifuged the filaments to discard the water. The procedure was repeated multiple times to record a second set of diffraction data.

**Transmission electron microscopy.** One droplet of Hfq peptide filaments was applied to glow-discharged 300 mesh carbon-coated copper grids for 1 min, washed with water, stained with 2% uranyl acetate (w/v) for 1 min, and dried under dark conditions. Samples were observed using a FEI CM120 transmission electron microscope at an accelerating voltage of 120 kV under TEM low-dose mode. TEM images were recorded using a Gatan USC1000 2k × 2k camera.

**SRCD.** For SRCD analyses, measurements and data collection were carried out on the DISCO beamline at Synchrotron SOLEIL, as described previously in Partouche et al.[65]. Protein secondary structure was determined with BeStSel software[66].

**ATR-FTIR.** Attenuated total reflection Fourier-transform infrared spectroscopy (ATR-FTIR) measurements were done on a Bruker Equinox55 spectrophotometer equipped with an MCT detector cooled with liquid nitrogen. 2 µL of the sample was deposited and dried with a nitrogen flow on a Specac diamond and 128 scans were averaged for sample and background. To remove most soluble or small oligomeric Hfq11 peptides (non-polymerized), the deposited sample was washed with a drop of 20 µL of water. The water was removed by pipetting without touching the sample and the sample was finally dried by nitrogen. For measurements of Hfq CTR amyloid structure on lipids, a drop of DOPG lipid solution at 0.5 mg/mL in chloroform (Avanti Polar Lipids, Alabaster, AL, USA) was first deposited on the IRE and dried with a nitrogen flux. 5 µL of D₂O from Cambridge Isotope was added on top and finally, 0.5 µL of assembled Hfq CTR at 20 mg/mL in D₂O was added. The system used was a Bruker Equinox55 purged with dry air and a diamond ATR device with a single reflection at an angle of 45° and closed with a golden gate chamber from Specac (Orpington, UK) to avoid evaporation and non-deuterated water vapor exchange. Spectra with a resolution of 4 cm$^{-1}$ were acquired every 90 min overnight.

**Solid-state NMR.** Solid-state NMR experiments were performed on a 14.1 T Bruker Biospin spectrometer equipped with 4 mm triple resonance ($^1$H, $^{13}$C, $^{15}$N) MAS probes. DSS (4,4-dimethyl-4-silapentane-1-sulphonic acid) was added to the sample for internal referencing. The spinning frequency was set to 11 kHz (4 mm rotor). The BCU temperature was set to 273 K for Hfq and 276 K for Hfq11, resulting in sample temperatures of 4 and 11 °C, respectively. All spectra were processed with Bruker TopSpin 4.2.0 and analyzed with the CcpNmr 2.4.2 software. The 2D $^{13}$C–$^{13}$C correlation spectrum was recorded using a $^1$H-$^{13}$C CP and a PDSD mixing time of 50 ms. Acquisition times were set to 10 ms and 15 ms in the indirect and the direct $^{13}$C dimension, respectively. 1D $^{13}$C-detected $^1$H–$^{13}$C CP and $^1$H–$^{13}$C INEPT (insensitive nuclei enhancement by polarization transfer)[67] spectra on Hfq11 and Hfq were both recorded with 20 ms acquisition time.

**Statistics and reproducibility.** Production and purification were performed and optimized for unlabeled and $^{13}$C, $^{15}$N-labeled Hfq samples resulting in protein samples with comparable purity. Synthetic Hfq11 fibril formation has been reported previously[21]. Multi-dimensional solid-state NMR experiments were recorded in experimental building blocks, encompassing approximately one day of experiment time, and added together.

**Reporting summary.** Further information on research design is available in the Nature Portfolio Reporting Summary linked to this article.

### Data availability
The datasets generated during and/or analyzed during the current study are available from the corresponding author on request.

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

## Acknowledgements

We acknowledge funding from the Center National de la Recherche Scientifique CNRS Momentum (BH), IdEx University of Bordeaux–Chaire d'installation (BH), the Région Nouvelle Aquitaine (BH, grant convention N◦ 2017-1R10305-00013031), the "Fondation de la Maison de la Chimie" (BH), the French National Research Agency (BH, grant No. ANR-19-CE13-0021), CNRS and CEA (VA). This study contributes to the IdEx Université Paris Cité ANR-18-IDEX-0001 (VA). The work has benefited from the Biophysical and Structural Chemistry Platform at IECB, CNRS UAR 3033, INSERM US001. We are grateful to Vincent Raussens (ULB, Bruxelles) for his help in FTIR analysis. SRCD experiments were performed on the "DISCO" beamline at SOLEIL Synchrotron, France (inhouse project:99230050). Financial support from the IR INFRANALYTICS FR2054 for conducting the research is gratefully acknowledged.

## Author contributions

M.B., D.M., B.K., E.M., A.G., and B.H. conducted the NMR, X-ray diffraction, and EM experiments and analyzed the data. F.W., J.W., and V.A. conducted the FTIR and SRCD experiments and analyzed the data. M.B. and D.M. performed protein expression and purification. V.A. and B.H. designed the project. J.W., F.W., B.K., V.A., and B.H. wrote the paper. All authors commented on the paper.

## Competing interests

The authors declare no competing interests.
