## [Peer Review File · Communications Biology]

Reviewers' comments:

Reviewer #1 (Remarks to the Author):

The main goal of this manuscript is to characterize the structure of Hfq C-terminal region using mainly solid state NMR as well as advanced biophysical methods including cyclotronic circular dichroism spectroscopy, FTIR and X-ray diffraction. The experiments are very well done and are rigorously analyzed. Overall this is a strong manuscript with interesting results that is well written. The presentation of the data in graphics is also excellent. The Results of the manuscript do advance the field, and raise some important new questions regarding the regulation of amyloid formation of the Hfq C-terminal region.

Comments:

1. For the Introduction section, consider including a descriptive figure showing the overall structure of the Hfq hexamer and the amino acid sequence of the protein in which the C-terminal disordered segment is highlighted.
2. The authors note that an eleven residue segment in the C-terminal disordered region has been identified as the minimal segment capable of assembling into fibrils but could they clarify whether the fibril forming segment could be longer in the context of the complete protein?
3. Material and Methods: Page 2, line 78. Glycerol: The "G" should be in lower case. NH₄Cl: The "4" should be in subscript.
4. Regarding X-ray diffraction, what exactly is meant by washing the peptide sample extensively using multiple centrifugation steps to record a second set of diffraction data?
5. Page 4, lines 44. It might be possible to assign the Asp versus Asn by lowering the pH to 3. Since this Asp is near the end of the amyloid, its side chain COO⁻ may be exposed enough to titrate to COOH. This would alter its ¹³Cβ chemical shift, whereas that of Asn would remain unaltered.
6. Page 4, lines 55-56, Would it not be more likely that the cluster of peaks at 65 ppm correspond to C-beta of Ser in the beta conformation and not C-alpha of Ser ? (this is properly shown in Fig. 2B).
7. Figure 1A. Can any information be gleaned from the ¹³CO region of this spectrum (180 - 170 ppm). What is the source of the weak peak at 110 - 100 ppm?
8. Figure 1, Panel D. The position of the parallel and anti-parallel labels should be moved next to the corresponding "arrow" schematic diagram. Two extra atomic structures should be added to represent the anti-parallel arrangement.
9. Figure 2, panel A. in the amino acid sequence: SAQNTSAQQDS, why are some residues labeled in blue and others in black?
10. Figure 3, panel A. What is the assignment and structural interpretation of the FTIR band at 1740 cm⁻¹?
11. Figure 3, panel B. What is the real precision of the % structure values obtained from CD. Are they really as small as tenths of a percent, as indicated in the table included in this figure?
12. Regarding the comparison of Sparta+ and experimental chemical shift values, in general the agreement seems to be good, but some peaks, esp. a Thr Cα predicted at 62 ppm but with an experimental value of about 58.5 ppm are poorly predicted. What could be the explanation for this case and others like it? Could this be the only Thr positioned in a beta strand in the structure? Could it

be a Thr close in space to an aromatic group? Also, is the Sparta+ algorithm temperature sensitive?

13. The authors compare the ssNMR spectra of the full length Hfq protein with that of the eleven residue segment and conclude that the conformation of the peptide is the same. Here, to further confirm this result, the authors could also record the ssNMR spectra of the hexameric Hfq protein without the C-terminal extension and show that the C-terminal segment's peaks are now missing.

14. Figure 5, page 8, legend. "grey" (British English). Check for consistency with American English versus British English spelling.

15. Page 5, line 73. "Ile" the "I" should be in lower case.

16. Discussion Section. The first three paragraphs of the Discussion are mostly dedicated to re-hashing the Introduction and Results section. Perhaps it could be substituted, at least in part, by a discussion the novel findings obtained here with the current state of the art in the Hfq field citing recent papers from other labs. This would highlight the relevance and impact of the results reported here.

In addition, these results raise new important questions, such as the basis of the stability of the amyloid fibril reported here and how its formation is regulated. The fact that the eleven residue sequence contains four residues that can be phosphorylated (three Ser and one Thr) and one residue whose charge state is sensitive to pH (Asp), already suggests a mechanism for biological regulation of amyloid formation.

Reviewer #2 (Remarks to the Author):

This manuscript by Berbon and colleagues, report a series of biophysical and structural studies on the full-length riboregulatory Hfq from *Escherichia coli* as well as studies on an 11-residue peptide that they hypothesised is responsible for the amyloid characteristics of the C-terminal region of Hfq. From their data, this 11-residue peptide shows strong β -strand features, which are the hallmarks of amyloid-like structures. The work is very interesting and well-presented. However, there remain some issues that the authors should address.

The authors claim the CTR and in particular the 11-residues within this region are responsible for multimerisation of Hfq molecules and these multimers might play a physiological role, perhaps in regulating Hfq activity. They also claim the CTR is important to protein stability. The question then is how do these Hfq multimers/" aggregates" disassemble to form functional full-length hexamers? It would seem that once multimerised, going back to monomeric hexamers would be thermodynamically unfavourable? This issue has to be clarified.

Is the order of the sequence critical to the β -rich amyloid structure? The authors should carry out SRCD analysis and x-ray diffraction on an 11-mer peptide with an E. coli sequence, in which the sequence content is the same, but scrambled.

There are other Hfq proteins with CTRs. Is the sequence of the 11-residue sequence that the authors have studied here conserved in other Hfq proteins from different genera/species? They should try to "Blast" this sequence against Hfq proteins from other bacteria to learn if there are the same or similar regions in these other riboregulators.

Reviewer #3 (Remarks to the Author):

This work by Berbon et al reports the formation of the beta-rich amyloid-like structure formed by the C-terminal region of Hfq protein, which can be maintained even with full-length Hfq. To demonstrate the filament assembly, the authors used several biophysical characterizations including solid state NMR, X-ray diffraction, infrared spectroscopy (IR) and circular dichroism (CD). While the finding that the C-terminal region of Hfq can assemble into filament structure is interesting from structure and protein assembly perspective, the biological relevance of this structure is largely lacking from the manuscript. I have several concerns listed below:

1. My major criticism is that the manuscript doesn't have any data to illustrate whether such amyloid structure exist in the cell at any conditions. So far, I am not aware of any study demonstrating such amyloid structure of Hfq, even though it's been shown the Hfq can form phase-separated condensate at specific cellular condition.
2. The assembly condition of Hfq in vitro for such amyloid structure is also not described in detail. The author mention that the filaments form in water, which is also not anywhere close to any intracellular condition. Proteins can simply aggregate without any buffer or salt. Under what concentration they can form filament is also not mentioned.
3. There are no Hfq hexamer controls provided in the biophysical characterizations for a rigorous comparison. It is unclear whether the signals are specific to the formation of amyloid structure or any assembly state compared to regular solution state of Hfq, which is a homo hexamer.
4. It would be very helpful for have a structure model to illustrate how the filament exactly looks, particularly for the full-length Hfq. It is hard to imagine how the core domain arranges when the C-terminal forms the amyloid structure.

Please find enclosed the revised version of our manuscript entitled "Hfq C-terminal region forms a β -rich amyloid-like motif without perturbing the N-terminal Sm-like structure". We now included one additional figure in the main text and five additional supplementary figures, as compared to the original submission, to answer reviewers' concerns. Please find below a point-by-point response to the reviewers' comments:

Response to Reviewer #1

1. For the Introduction section, consider including a descriptive figure showing the overall structure of the Hfq hexamer and the amino acid sequence of the protein in which the C-terminal disordered segment is highlighted.

We thank the reviewer for this idea; the suggested figure has now been added in the introduction.

2. The authors note that an eleven residue segment in the C-terminal disordered region has been indentified as the minimal segment capable of assembling into fibrils but could they clarify whether the fibril forming segment could be longer in the context of the complete protein?

The referee's comment is correct and the amyloid region should be longer in the context of the protein. We have found the 11-aa region to reflect the minimum region needed to form the amyloid assembly and therefore constitutes the nucleation region, but we also showed that other residues in the 38 aa residues CTR region contribute to the amyloid assembly¹. These residues include the region from S₆₅ to G₇₈². Note that the region S₆₅RPVSHHSNAGGGT₈₀ by itself is not able to polymerize, indicating the nucleation region includes this 11 aa region (see Table S1 in (Malabirade, Morgado-Brajones et al. 2017)¹). This has now been indicated in the discussion.

3. Material and Methods: Page 2, line 78. Glycerol: The "G" should be in lower case. NH4Cl: The "4" should be in subscript.

This has been corrected.

4. Regarding X-ray diffraction, what exactly is meant by washing the peptide sample extensively using multiple centrifugation steps to record a second set of diffraction data?

After performing the first X-ray diffraction experiments, we have used the remaining sample of Hfq11 filaments, resuspended the filaments in H₂O and centrifuged to discard the H₂O. We have repeated this procedure multiple times. We have modified the sentence in the manuscript accordingly.

5. Page 4, lines 44. It might be possible to assign the Asp versus Asn by lowering the pH to 3. Since this Asp is near the end of the amyloid, its side chain COO- may be exposed enough to titrate to COOH. This would alter its 13Cbeta chemical shift, whereas that of Asn would remain unaltered.

We thank the reviewer for his valid comment. Since we observe two resonance peaks for Asp and Asn and only two residues of these types are present in the peptide, we did not seek to assign Asp vs. Asn.

6. Page 4, lines 55-56, *Would it not be more likely that the cluster of peaks at 65 ppm correspond to C-beta of Ser in the beta conformation and not C-alpha of Ser ? (this is properly shown in Fig. 2B).*

We have corrected the error.

7. Figure 1A. *Can any information be gleaned from the ^{13}CO region of this spectrum (180 - 170 ppm). What is the source of the weak peak at 110 – 100 ppm?*

Because the resonance peaks are crowded in this region, no residue-specific assignment can be performed and the information content obtained at the signal to noise of the unlabelled peptide is therefore limited. Still, we agree that the information can still be related to the secondary structure content and we have now added an analysis in the Figure 2 (now 3) Panel A, an additional Figure S4 (see below), and a descriptive text. We thank the reviewer for this remark.

The resonance peaks at 110 – 100 ppm correspond to spinning sidebands of the C=O resonance peaks (around 172-178 ppm) due to the MAS frequency of 11 kHz, corresponding to approx. 73 ppm. We have added a sentence for clarification.

8. Figure 1, Panel D. *The position of the parallel and anti-parallel labels should be moved next to the cooresponding “arrow” schmatic diagram. Two extra atomic structures should be added to represent the anti-parallel arrangement.*

We have clarified the Figure 1, panel D. We show the monomeric peptide when modelled for a parallel vs. antiparallel conformation because without atomic restraints we do not want to mislead the reader with an atomic structure model for the filaments by showing an intermolecular arrangement.

9. Figure 2, panel A. *in the amino acid sequence: SAQNTSAQQDS, why are some residues labeled in blue and others in black?*

We have coloured all residues in black now.

10. Figure 3, panel A. *What is the assignment and structural interpretation of the FTIR band at 1740 cm^{-1} ?*

The band observed around 1740 cm^{-1} corresponds to the C=O stretching vibration of an ester. As active esters are often used for solid phase peptide synthesis, we suspect this band coming from a slight contamination occurring during peptide synthesis³. This ester band is outside of the amide I used for the characterization of the secondary structure and therefore does not interfere with our results. We have added an explanatory sentence in the manuscript section 3.2.

11. Figure 3, panel B. What is the real precision of the % structure values obtained from CD. Are they really as small as tenths of a percent, as indicated in the table included in this figure?

BeStSel fits the experimental CD spectra by a linear combination of fixed basis components in order to obtain the proportion of eight structural elements, which are reduced to five secondary structures including an α -helix, antiparallel and parallel β -sheet, turn and others. The error of the structure estimation given to the decimal correlates with the NMRSD (normalised root mean square deviation) of the CD fitting, corresponding to an NMRSD of 0.058 for the analysis presented here. Indeed, we have presented the secondary structure content including the decimals mostly because adding up the secondary structure elements should sum up to 100%. In the legend of Figure 3, we have added the NMRSD for the spectral fitting.

12. Regarding the comparison of Sparta+ and experimental chemical shift values, in general the agreement seems to be good, but some peaks, esp. a Thr Calpha predicted at 62 ppm but with an experimental value of about 58.5 ppm are poorly predicted. What could be the explanation for this case and others like it? Could this be the only Thr positioned in a beta strand in the structure? Could it be a Thr close in space to an aromatic group? Also, is the Sparta+ algorithm temperature sensitive?

The reviewer is raising an important point. The structural data we present in the 2D ^{13}C - ^{13}C ssNMR spectrum is atomic resolution data on the full-length protein in the filamentous form. The predicted peaks have been calculated by SPARTA+ on a monomeric structure (PDB code 4RCB). We therefore don't expect the predicted values to correspond exactly to the observed chemical shifts; the similar chemical shift dispersion, the high resolution as well as the distribution of amino acids in a rigid well-folded conformation, described in the manuscript, lead us to the conclusion that the globular domain should be present in its conserved monomer structure in the full-length Hfq filaments. The expected error on the chemical shift prediction by SPARTA+ is in the range of ≈ 1 ppm for $\text{C}\alpha$ and ≈ 1.3 ppm for $\text{C}\beta$, as evaluated in Shen et al. 2010 on chemical shift predictions of 11 proteins⁴. We have now performed SPARTA+ chemical shift predictions on the hexameric Hfq (PDB code 3QHS) to compare the chemical shift predictions of the monomeric vs. the hexameric structure and evaluate them in comparison with our chemical shift data on full-length Hfq in filaments. For the hexameric assembly the predicted signals of the two Thr residues coincide more with the central Thr peak around 62/70 ppm (see below and new Figure S5). We have added a paragraph in section 3.3. to explain the issue.

Figure: ^{13}C - ^{13}C PDSO ssNMR spectrum of Hfq fibers, as in Figure 5 in the manuscript. Resonance peaks (violet) of Thr $\text{C}\alpha$ - $\text{C}\beta$ correlations, predicted for the Hfq hexamer structure (PDB code: 3QHS, chemical shift predictions from Sparta+), shown on the selected spectral region of Thr signals. Thr residues are highlighted in red on the monomer (grey, PDB code 4RCB) and aligned hexamer (rainbow colour for the aligned hexamer and dark grey for the second hexamer in the asymmetric unit, PDB code 3QHS).

13. The authors compare the ssNMR spectra of the full length Hfq protein with that of the eleven residue segment and conclude that the conformation of the peptide is the same. Here, to further confirm this result, the authors could also record the ssNMR spectra of the hexameric Hfq protein without the C-terminal extension and show that the C-terminal segment's peaks are now missing.

This experiment would be interesting but Hfq hexamer devoid of C-terminal does not self-assemble in fibers and thus cannot be analysed by ssNMR (see also our answer to referee 3 point 4). Truncated Hfq globular domain and full-length Hfq have been analysed in the soluble form by solution NMR⁵ (reference 9 of the manuscript) but the chemical shifts have unfortunately not been deposited in the dedicated BMRB database. When comparing solution NMR between the globular domain and the full-length Hfq in solution, significant differences in the resolution, as a consequence of the higher molecular weight of the hexameric complex and possibly further intermolecular interactions, and of the detected chemical shifts have been observed. This underlines that using chemical shift predictions by SPARTA+ seems an adequate solution to discuss the chemical shifts for our purpose. We have now added a paragraph in the manuscript discussion to provide a more detailed view on the comparison between solution and solid-state NMR.

14. Figure 5, page 8, legend. "grey" (British English). Check for consistency with American English versus British English spelling.

We checked to follow British English spelling throughout the document.

15. Page 5, line 73. "Ile" the "I" should be in lower case.

Has been done. We thank the reviewer for his attentive reading.

16. Discussion Section. The first three paragraphs of the Discussion are mostly dedicated to re-hashing the Introduction and Results section. Perhaps it could be substituted, at least in part, by a discussion the novel findings obtained here with the current state of the art in the Hfq field citing recent papers from other labs. This would highlight the relevance and impact of the results reported here.

We have extended the discussion by further including the suggested modifications and hope to satisfy the reviewers' expectations.

In addition, these results raise new important questions, such as the basis of the stability of the amyloid fibril reported here and how its formation is regulated. The fact that the eleven residue sequence contains four residues that can be phosphorylated (three Ser and one Thr) and one residue whose charge state is sensitive to pH (Asp), already suggests a mechanism for biological regulation of amyloid formation.

We now suggest this possibility in the discussion section. We also mention that a careful MS analysis has been done few years ago by colleagues and that they identify a post-translational modification in Hfq, an oxidised lipid⁶. They do not evidence a phosphorylation at this time but it could be the condition of growth needed to allow the phosphorylation could be different.

Reviewer #2 (Remarks to the Author):

This manuscript by Berbon and colleagues, report a series of biophysical and structural studies on the full-length riboregulatory Hfq from Escherichia coli as well as studies on an 11-residue peptide that they hypothesised is responsible for the amyloid characteristics of the C-terminal region of Hfq. From their data, this 11-residue peptide shows strong β -strand features, which are the hallmarks of amyloid-like structures. The work is very interesting and well-presented. However, there remain some issues that the authors should address.

The authors claim the CTR and in particular the 11-residues within this region are responsible for multimerisation of Hfq molecules and these multimers might play a physiological role, perhaps in regulating Hfq activity. They also claim the CTR is important to protein stability. The question then is how do these Hfq multimers/" aggregates" disassemble to form functional full-length hexamers? It would seem that once multimerised, going back to monomeric hexamers would be thermodynamically unfavourable? This issue has to be clarified.

We thank the reviewer for raising this question. Indeed, we know that cellular actors influence this equilibrium and that for instance lipids found in biological membrane allow the disassembly of amyloid CTR region. Indeed, it is known that Hfq interact with inner membrane *in vitro* and *in vivo*⁷¹. This effect of lipid can be observed on the figure below where pre-polymerized Hfq-CTR was exposed to DOPG lipids. We clearly see that the peak at 1610 cm^{-1} disappears, evidencing the disassembly of the amyloid structure, indicating that multimerised amyloid can go back to a hexameric form in the presence of certain cellular components. Note that the SRCD spectrum also confirms that the CTR is not amyloid-like after the interaction with membrane lipids. This has now been added in section 3.3, the discussion and in the Figure S6 (see figure below). We have further extended the discussion towards including recent reports on the interactions with other cellular partners and the possible regulative mechanisms, such as post-translational modifications including acylation⁶.

Figure: FTIR spectra obtained after adding Hfq-CTR fibrils on DOPG lipids. The spectral pattern changes over 12 hours from blue to cyan color. The decreasing signal at 1610 cm^{-1} indicates that Hfq CTR fibrils disassemble into a non-amyloid structure.

*Is the order of the sequence critical to the β -rich amyloid structure? The authors should carry out SRCD analysis and x-ray diffraction on an 11-mer peptide with an *E. coli* sequence, in which the sequence content is the same, but scrambled.*

Indeed, we believe this region of 11 aa might form a “steric zipper”-like assembly, *i.e.* pairs of self-complementary β -sheets formed by short sequences found in amyloids^{8,9}:

In these “zippers”, pairs of asparagines and glutamines can form hydrogen bonds along the fibrils.

This possibility is strongly supported by the fact that removing the (Q₉₀N₉₁) and (Q₉₅Q₉₆) sequences abolishes the self-assembly¹. Furthermore, we also observed that mutating the Serines S₈₈, S₉₃ or S₉₈ for an alanine in the 11-mer region also abolishes the self-assembly. Thus the order and the nature of the sequence is important for the self-assembly². This has now been indicated in the text.

There are other Hfq proteins with CTRs. Is the sequence of the 11-residue sequence that the authors have studied here conserved in other Hfq proteins from different genera/species? They should try to “Blast” this sequence against Hfq proteins from other bacteria to learn if there are the same or similar regions in these other riboregulators.

We thank the referee for his suggestion. Indeed, this type of analysis has already been published, such as in Sun and Wartell, 2002¹⁰ and Turbant, O. et al. 2021². The reports show that there are two variants of Hfq: (i) Hfq proteins of γ - and β -proteobacteria that have an extended C-terminus; (ii) other Hfq lacking a CTR, including those of Gram-positive bacteria such as *S. aureus* Hfq. Obviously, bacteria without an extended CTR, don’t present such a sequence. Conversely, we noticed that Hfq with an extended CTR such as that of *Yersinia* sometimes have a region of 11 aa very similar to that of *E. coli* Hfq. But this is not always the

case. For instance, Hfq of *P. brassicacearum* has a CTR but lack an equivalent region of 11 aa similar to that of *E. coli* Hfq. Since in the context of our manuscript, the introduction of the C-terminal conservation contributes important information, we performed additional “Blast” predictions and added a paragraph in the Discussion section and Figure S7.

Reviewer #3 (Remarks to the Author):

This work by Berbon et al reports the formation of the beta-rich amyloid-like structure formed by the C-terminal region of Hfq protein, which can be maintained even with full-length Hfq. To demonstrate the filament assembly, the authors used several biophysical characterizations including solid state NMR, X-ray diffraction, infrared spectroscopy (IR) and circular dichroism (CD). While the finding that the C-terminal region of Hfq can assemble into filament structure is interesting from structure and protein assembly perspective, the biological relevance of this structure is largely lacking from the manuscript. I have several concerns listed below:

1. My major criticism is that the manuscript doesn't have any data to illustrate whether such amyloid structure exist in the cell at any conditions. So far, I am not aware of any study demonstrating such amyloid structure of Hfq, even though it's been shown the Hfq can form phase-separated condense at specific cellular condition.

We thank the reviewer for this comment. Indeed, we and others have recently published three reports addressing this aspect. In Partouche et al. 2019 we show, using FTIR spectroscopy on whole cells, that Hfq forms an amyloid structure *in cellulo*¹¹. Cossa et al. J Struct. Biol. 2022¹² and Beaufay et al. Sci. Adv. 2021¹³ are recent papers showing that the C-terminal region of Hfq influences DNA compaction *in vivo*, for example using mutated forms of Hfq and cryo-soft X-ray crystallography¹². In addition to driving phase separation *in vivo*, Beaufay et al. Sci. Adv. 2021¹³ report the formation of elongated high MW multimers, we therefore have strong evidence that Hfq forms this type of structure *in vitro* and *in vivo*. We agree however that high MW assembled structures may be less stably formed in most cellular conditions *in vivo*. We have added a paragraph addressing this issue in the discussion. Note that unfortunately for now we cannot produce anti-Hfq-oligomer specific antibodies, even if we tried to produce them (but were not successful).

2. The assembly condition of Hfq in vitro for such amyloid structure is also not described in detail. The author mention that the filaments form in water, which is also not anywhere close to any intracellular condition. Proteins can simply aggregate without any buffer or salt. Under what concentration they can form filament is also not mentioned.

We thank the reviewer for raising the issue. Hfq assembled spontaneously at a concentration of 4 mg/ml at room temperature under mild agitation in 50 mM Tris-HCl, 50 mM NH₄Cl, 10 % Glycerol, pH = 7.4. We have now modified the section “Protein expression, purification and assembly” in Materials and Methods accordingly, and added a refined legend to Fig. S1 to provide more details for this section.

Proteins and peptides tend to aggregate when assembled under certain conditions. However, we know that the assembled structures we observe are not simply unstructured aggregates because we observe solid-state NMR linewidths of approx. 100-150 Hz, as is not the case for example in unstructured or less well-ordered aggregates such as heated filaments or less ordered aggregates (see for example ¹³C-¹³C correlation spectra in A β aggregates in Fig. 5 in Tay et al. 2013¹⁴, or heated prion protein in Fig. 5e in Loquet et al. 2009¹⁵).

3. *There are no Hfq hexamer controls provided in the biophysical characterizations for a rigorous comparison. It is unclear whether the signals are specific to the formation of amyloid structure or any assembly state compared to regular solution state of Hfq, which is a homo hexamer.*

Because Hfq hexamers including only the globular domain do not assemble, they cannot be examined by solid-state NMR, while the assembly of Hfq monomers into hexamers leads to line broadening in solution NMR⁵, usually used for proteins with a molecular weight < 40kDa, precluding its application to the hexamers. The here-performed solid-state NMR proton-driven spin diffusion experiments were performed with an initial ¹H-¹³C cross-polarisation and would therefore only record signals originating from residues in a rigid assembled regime. If the filaments would assemble through interactions between the globular domains of the hexamers, including a floppy intrinsically disordered C-terminal, the solid-state NMR spectra would only contain peaks originating from residues in the globular domain, such as two peaks for Thr 49 and Thr 61 as indicated by the peak predictions shown in Fig. 5a and the now added Fig. S5. The signal intensity and resolution of the detected Thr signals suggests the presence of 4 Thr residues in a highly structured β -strand conformation. Since the spin systems of isoleucines, exclusively present and well distributed in the globular domain, are well-resolved and only present in one set, we can exclude global conformational doubling. Furthermore, increase in β -sheet content has been shown for Hfq in filamentous assemblies¹⁶ and we show that the 11-residue peptide assembles into amyloid filaments with one set of Thr α -C β correlations precisely located at the Thr peak positions of the full-length Hfq, indicative of a peak originating from the same Thr residue. We can therefore conclude that our data suggest amyloid-like interactions to occur in the filamentous assemblies of full-length Hfq filaments, including the minimal 11-residue motif, possibly modulating RNA/DNA interactions and phase separation properties.

4. *It would be very helpful for have a structure model to illustrate how the filament exactly looks, particularly for the full-length Hfq. It is hard to imagine how the core domain arranges when the C-terminal forms the amyloid structure.*

We thank the reviewer for this suggestion. We have included an explanatory text in the manuscript discussion and a Figure S8 to propose how amyloid-like interactions could help regulating Hfq functions. Further investigations are required to provide more detailed insights into the molecular arrangements at the atomic level.

References

1. Malabirade, A. *et al.* Membrane association of the bacterial riboregulator Hfq and functional perspectives. *Sci Rep* **7**, 10724 (2017).
2. Turbant, F. *et al.* Identification and characterization of the Hfq bacterial amyloid region DNA interactions. *BBA Advances* **1**, 100029 (2021).
3. Bodanszky, M. & Fagan, D. T. ACTIVE ESTERS IN THE FORMATION OF ESTER BONDS BETWEEN AMINO ACIDS AND POLYMERIC SUPPORTS. *International Journal of Peptide and Protein Research* **10**, 375–379 (2009).
4. Shen, Y. & Bax, A. Prediction of Xaa-Pro peptide bond conformation from sequence and chemical shifts. *J Biomol NMR* **46**, 199–204 (2010).
5. Beich-Frandsen, M. *et al.* Structural insights into the dynamics and function of the C-terminus of the E. coli RNA chaperone Hfq. *Nucleic Acids Research* **39**, 4900–4915 (2011).
6. Obregon, K. A., Hoch, C. T. & Sukhodolets, M. V. Sm-like protein Hfq: Composition

of the native complex, modifications, and interactions. *Biochimica et Biophysica Acta (BBA) - Proteins and Proteomics* **1854**, 950–966 (2015).

7. Diestra, E., Cayrol, B., Arluison, V. & Risco, C. Cellular Electron Microscopy Imaging Reveals the Localization of the Hfq Protein Close to the Bacterial Membrane. *PLoS ONE* **4**, e8301 (2009).

8. Sawaya, M. R. *et al.* Atomic structures of amyloid cross- β spines reveal varied steric zippers. *Nature* **447**, 453–457 (2007).

9. Park, J., Kahng, B. & Hwang, W. Thermodynamic Selection of Steric Zipper Patterns in the Amyloid Cross- β Spine. *PLoS Comput Biol* **5**, e1000492 (2009).

10. Sun, X. Predicted structure and phyletic distribution of the RNA-binding protein Hfq. *Nucleic Acids Research* **30**, 3662–3671 (2002).

11. Partouche, D. *et al.* In Situ Characterization of Hfq Bacterial Amyloid: A Fourier-Transform Infrared Spectroscopy Study. *Pathogens* **8**, 36 (2019).

12. Cossa, A. *et al.* Cryo soft X-ray tomography to explore Escherichia coli nucleoid remodeling by Hfq master regulator. *Journal of Structural Biology* **214**, 107912 (2022).

13. Beaufay, F. *et al.* Polyphosphate drives bacterial heterochromatin formation. *Sci. Adv.* **7**, eabk0233 (2021).

14. Tay, W. M., Huang, D., Rosenberry, T. L. & Paravastu, A. K. The Alzheimer's Amyloid- β (1–42) Peptide Forms Off-Pathway Oligomers and Fibrils That Are Distinguished Structurally by Intermolecular Organization. *Journal of Molecular Biology* **425**, 2494–2508 (2013).

15. Loquet, A. *et al.* Prion fibrils of Ure2p assembled under physiological conditions contain highly ordered, natively folded modules. *J Mol Biol* **394**, 108–18 (2009).

16. Arluison, V. *et al.* Three-dimensional Structures of Fibrillar Sm Proteins: Hfq and Other Sm-like Proteins. *Journal of Molecular Biology* **356**, 86–96 (2006).

REVIEWERS' COMMENTS:

Reviewer #1 (Remarks to the Author):

The authors have done a fine job addressing all my concerns. This is an excellent paper and a valuable addition to the literature.

Reviewer #2 (Remarks to the Author):

This revised manuscript has addressed the major concerns of this reviewer. The authors provide thorough and thought-provoking data that support the importance of this C-terminal 11mer peptide sequence in "amyloid" nucleation via the production of *β* strands/*β* sheets, which of course could play a role in one or more of the functions of Hfq in the cell.

Reviewer #3 (Remarks to the Author):

The authors have addressed my technical concerns in the revised manuscript and their responses to the reviewers. While I am not fully convinced that Hfq protein form amyloid structures in the cell, the in vitro data itself presented in the current manuscript could be interesting from a structural biology perspective. I respect the editor's final decision.

Please find enclosed the final version of our manuscript entitled "Hfq C-terminal region forms a β -rich amyloid-like motif without perturbing the N-terminal Sm-like structure". We have adjusted the manuscript to the guidelines of *Communications Biology* and therefore needed to change the order of the text sections and the order in the reference list. Please find below a response to the reviewers' comments.

Response to Reviewer #1

The authors have done a fine job addressing all my concerns. This is an excellent paper and a valuable addition to the literature.

We would like to thank the reviewer for his valuable ideas for the manuscript at the first stage and his positive evaluation of the manuscript after revision.

Response to Reviewer #2

This revised manuscript has addressed the major concerns of this reviewer. The authors provide thorough and thought-provoking data that support the importance of this C-terminal 11mer peptide sequence in "amyloid" nucleation via the production of β strands/ β sheets, which of course could play a role in one or more of the functions of Hfq in the cell.

We thank the reviewer for his remark and we are glad to satisfy his expectations for an added thought-provoking value to the scientific community.

Response to Reviewer #2

The authors have addressed my technical concerns in the revised manuscript and their responses to the reviewers. While I am not fully convinced that Hfq protein form amyloid structures in the cell, the in vitro data itself presented in the current manuscript could be interesting from a structural biology perspective. I respect the editor's final decision.

We thank the reviewer for having invested the time to evaluate our manuscript and for improving the quality of the manuscript.